# Potential impact, costs, and benefits of population-wide screening interventions for tuberculosis in Viet Nam: A mathematical modelling study

Alvaro Schwalb[1,2,3*], Katherine C. Horton[1,2], Jon C. Emery[1,2], Martin J. Harker[1,2,4], Lara Goscé[1,2], Lara D. Veeken[5], Frances L. Garden[6,7], Hai Viet Nguyen[8], Thu-Anh Nguyen[9,10,11,12], Khanh Luu Boi[12], Frank Cobelens[13,14], Greg J. Fox[10,11,12], Van Luong Dinh[15,16], Hoa Binh Nguyen[15,16], Guy B. Marks[6,12,17,18], Rein M. G. J. Houben[1,2]

**1** TB Modelling Group, TB Centre, London School of Hygiene and Tropical Medicine, London, United Kingdom, **2** Department of Infectious Disease Epidemiology, London School of Hygiene and Tropical Medicine, London, United Kingdom, **3** Instituto de Medicina Tropical Alexander von Humboldt, Universidad Peruana Cayetano Heredia, Lima, Peru, **4** Global Health Economics Centre, London School of Hygiene and Tropical Medicine, London, United Kingdom, **5** Department of Internal Medicine and Radboud Community for Infectious Diseases, Radboud University Medical Center, Nijmegen, the Netherlands, **6** South West Sydney Clinical Campuses, University of New South Wales, Sydney, Australia, **7** Ingham Institute of Applied Medical Research, Sydney, Australia, **8** Ministry of Health, Hanoi, Viet Nam, **9** The University of Sydney Vietnam Institute, Ho Chi Minh City, Viet Nam, **10** Faculty of Medicine and Health, University of Sydney, Sydney, Australia, **11** The University of Sydney Institute for Infectious Diseases, Sydney, Australia, **12** Woolcock Institute of Medical Research, Sydney, Australia, **13** Department of Global Health, Amsterdam University Medical Centers, University of Amsterdam, Amsterdam, The Netherlands, **14** Amsterdam Institute for Global Health and Development, Amsterdam, The Netherlands, **15** National Lung Hospital, National Tuberculosis Control Programme, Hanoi, Viet Nam, **16** Hanoi Medical University, Hanoi, Viet Nam, **17** School of Clinical Medicine, University of New South Wales, Sydney, Australia, **18** Burnet Institute, Melbourne, Australia

* alvaro.schwalb@lshtm.ac.uk

## Abstract

Population-wide screening may accelerate the decline of tuberculosis (TB) incidence, but the optimal screening algorithm and duration must weigh resource considerations. We calibrated a deterministic transmission model to TB epidemiology in Viet Nam. We simulated three population-wide screening algorithms from 2025: sputum nucleic acid amplification tests (NAAT, Xpert MTB/RIF Ultra) only; chest radiography (CXR) followed by NAAT; and CXR-only without microbiological confirmation. We determined the annual screening rounds required to reduce pulmonary TB prevalence below 50 per 100,000 people. Cost-effectiveness was assessed using incremental cost-effectiveness ratios (ICERs), representing the additional costs (in US$) per disability-adjusted life year (DALY) averted compared to business-as-usual by 2050. Additionally, we evaluated the impact of NAAT cartridges costing US$1 each. NAAT-based algorithms required at least six rounds to reach the prevalence threshold, while CXR-only required three. NAAT-only achieved a prevalence reduction

**Data availability statement:** Data and analysis code is available at GitHub (https://github.com/aschwalbc/ACF-VN).

**Funding:** This work was supported by the European Research Council [grant number 757699 to AS, JCE, KCH, and RMGJH]. KCH is also supported by UK FCDO (Leaving no-one behind: transforming gendered pathways to health for TB). This research has been partially funded by UK Aid from the UK government (to KCH); however, the views expressed do not necessarily reflect the UK government's official policies. GJF was supported by a Leadership Fellowship from the Australian National Health and Medical Research Council. The funders had no role in study design, data collection and analysis, decision to publish, or preparation of the manuscript.

**Competing interests:** The authors have declared that no competing interests exist.

consistent with the ACT3 trial after three rounds. The CXR+NAAT algorithm averted 4.29m DALYs (95%UI:2.86-6.14) at US$225 (95%UI:85–520) per DALY averted compared with business-as-usual. The front-loaded investment of US$161m (95%UI:111–224) annually during the intervention resulted in average annual cost savings of US$12.7m (95%UI:6.7-21.4) up to 2050 compared to the business-as-usual counterfactual. Reducing the cost of NAAT to US$1 led to a 50% and 15% reduction in budget impact and a 63% and 26% reduction in the estimated ICER for the NAAT-only and CXR+NAAT algorithms, respectively. In Viet Nam, population-wide screening could achieve ambitious policy goals. Substantial front-loaded investment is immediately followed by persistent cost savings and could be further offset by more affordable NAATs.

## Introduction

Despite a slight decline in tuberculosis (TB) incidence of approximately 2% over the past decade, an estimated 10 million people still fall ill with TB each year [1]. Worldwide, the conventional approach to TB prevention and care is passive detection, where diagnosis and treatment are only provided to individuals with symptoms who seek and receive healthcare [2–4]. This approach results in a large gap of undiagnosed individuals [1], as not everyone with TB experiences symptoms or is able to access care [5]. Furthermore, due to the long duration and undulating pattern of TB, it also results in onward transmission before diagnosis and treatment [6]. Thus, in order to achieve ambitious *End TB Strategy* targets [7], relying solely on passive detection is insufficient [3,6]. In contrast, population-wide screening aims to interrupt transmission by finding and treating individuals with TB disease who otherwise would be diagnosed later or not at all through the usual patient-initiated pathway [2,8,9]. Theoretically, when implemented intensively and consistently over multiple years, this approach could reduce TB prevalence and incidence by providing earlier treatment, shortening the period of infectiousness and breaking the chain of transmission [10].

Population-wide screening is not a novel approach; it has previously been employed in countries now considered to have a low TB burden [10–15]. Screening campaigns using mobile chest radiography (CXR) units were conducted as early as the 1930s, with remarkable results across different settings [16]. More recently, the ACT3 trial, conducted in Viet Nam from 2014 to 2018, implemented annual, community-wide screening over three years, using a symptom-agnostic approach similar to historic screening efforts [16]. Notably, the trial demonstrated a significant reduction in the prevalence of pulmonary TB in the communities where screening was employed, compared with those utilising routine, passive detection methods alone [17]. It also observed a 57% reduction in incident TB episodes over three successive annual cohorts [9]. This outcome provides contemporary evidence of the tangible impact of population-wide screening, highlighting the need for proactive measures into current approaches to TB prevention and care.

Although population-wide screening shows evident promise, its implementation as a central component of TB elimination strategies in high TB burden countries like Viet Nam remains under debate. A major challenge lies in determining the optimal way to implement these strategies—including the ideal duration and frequency of repeated screening, and the most appropriate screening algorithm. Resolving these factors is crucial to ascertaining the most effective approach to reducing the TB burden, both in terms of incident TB and deaths averted. Furthermore, population-wide screening requires significant investment, thus scaling up its implementation must consider the substantial front-loaded costs, including financial and human resources [18–20]. We sought to identify the most effective algorithm and duration of population-wide screening necessary to achieve a significant reduction in TB prevalence in Viet Nam, while weighing the short-term economic costs with long-term savings of reduced incident TB episodes.

## Methods

### Model structure

Previous modelling work has emphasised the need for enhanced TB screening and diagnosis of individuals [21–23]; nonetheless, it did not fully account for the spectrum of TB disease and the nature of transmission from individuals with asymptomatic TB [21,24]. To address this, we developed a deterministic transmission model of TB natural history that incorporates recent advances in quantifying the spectrum of TB disease [23,25], including self-clearance of *Mycobacterium tuberculosis* (*Mtb*) infection and the relative contribution of asymptomatic TB to transmission [26,27]. We represented the natural history of TB using nine compartments, reflecting the progression of *Mtb* infection along a spectrum from susceptibility to various states for disease and treatment (**Fig 1**). The three disease states were defined as follows: (i) non-infectious TB, representing individuals with inflammatory pathology in the absence of bacteriological evidence of TB disease and infectiousness, regardless of symptoms; (ii) asymptomatic TB, representing individuals with bacteriological evidence of TB disease who do not report symptoms during screening; and (iii) symptomatic TB, representing individuals with bacteriological evidence of TB disease who do report symptoms during screening. The transition from non-infectious to asymptomatic TB was defined by the presence of bacteriological evidence of disease, such as a positive

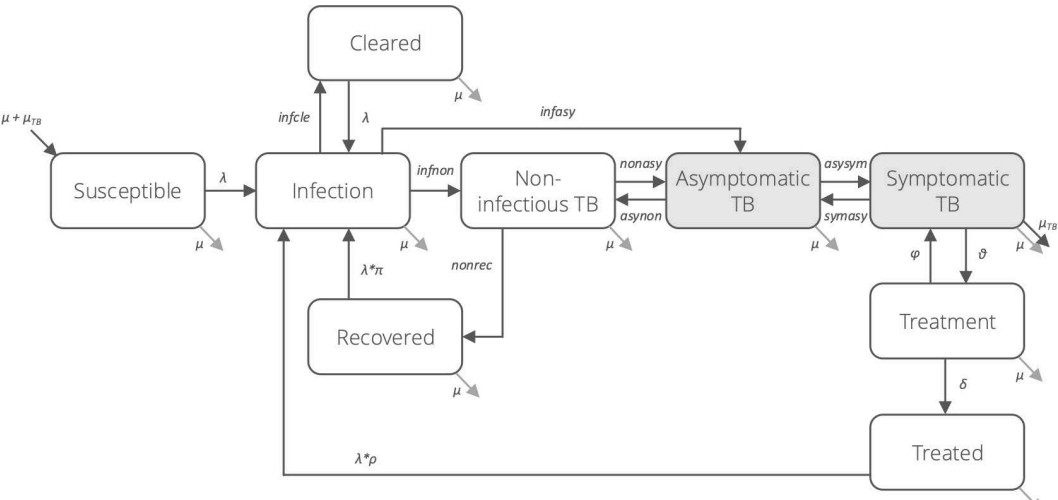

**Fig 1. TB natural history model structure.** A closed population, compartmental model of tuberculosis natural history, adapting features of previously published models [21,24]. The model is depicted using nine compartments across the spectrum of TB. Shaded compartments represent those that contribute to transmission. The force of infection (depicted with λ) depends upon the contact parameter and the prevalence of infectious disease (i.e., asymptomatic and symptomatic TB), adjusting for the relative infectiousness of asymptomatic TB.

sputum smear, nucleic acid amplification test (NAAT), or culture, indicating infectiousness; the transition from asymptomatic to symptomatic TB was determined by the presence of self-reported symptoms. The disease state classification was informed by the ICE-TB framework, and naming follows current World Health Organization (WHO) definitions [28]. Additional model states, parameterisation, and model equations are described in detail in S1-S2 Text. The model was constructed using R v4.2.3 for statistical computing and graphics [29].

## Model calibration

We calibrated the model to TB epidemiological data from Viet Nam using history matching with emulation, facilitated by the *hmer* R package (further details in the S3 Text) [23,30–33]. Calibration targets included TB prevalence, TB mortality rate, TB notification rate, and the proportion of prevalent infectious disease that is asymptomatic, each set for at least two distinct years (S1 Table) [16]. Ranges for priors and sources are shown in S2 Table. Posterior parameter sets are presented as median values with corresponding 95% uncertainty intervals (95% UI), calculated as the 2.5th to 97.5th percentiles of the parameter sets, to quantify uncertainty.

## Interventions

We simulated the population impact of different screening algorithms and durations as compared with the business-as-usual (BAU) counterfactual (see details in S4 Text). In the counterfactual scenario, we assumed that TB trends would follow the calibration trajectory, reflecting the ongoing provision of TB treatment and prevention services through routine passive detection, including a limited amount of individual or high-risk group-focused screening interventions (i.e., contact tracing, HIV screening) currently offered by the National TB Programme (NTP) of Viet Nam. Compared with the BAU baseline, we evaluated three population-wide screening intervention algorithms, all implemented regardless of whether individuals reported symptoms: (i) using an NAAT (Xpert MTB/RIF Ultra) only, (ii) a two-step approach using digital CXR with computer-aided diagnosis software interpretation, followed by NAAT for those with imaging abnormalities and (iii) using CXR only. NAAT-based algorithms were performed upon expectorated sputum. For all three algorithms, a positive screen led to treatment initiation, with no additional bacteriological test or clinical assessment performed prior to treatment. We selected 2025 as the earliest year when population-wide screening interventions could be implemented and assumed the entire adult population of Viet Nam would be eligible for screening. The population was uniformly screened across all model states (except for those undergoing TB treatment), with no increased probability of screening for individuals with TB disease. Additionally, we made the simplifying assumption that sputum production is unevenly distributed by TB status: all individuals with infectious TB (i.e., asymptomatic and symptomatic) were assumed to be able to provide an analysable sample, while only 60% of those with other model states would be able to do so. An exception was made under the CXR+NAAT algorithm, where we assumed that all individuals who screened positive to CXR would also be able to provide sputum, given radiological evidence of disease.

## Probability of a positive test

Screening was implemented based on the probability of a positive test for each model state according to the screening tool used; for the two-step algorithm, these probabilities were multiplied (**Table 1** and S3 Table). Probabilities were independently sampled from uniform distributions for each model run.

## Epidemiological outcomes

We sought to determine the number of repeated annual rounds of population-wide screening needed for the TB prevalence to fall below 50 per 100,000 people using each algorithm. We used the median value of the model outputs and considered the threshold met if it fell below or within 10% above the target. Full uncertainty intervals were presented for all

**Table 1. Probability of a positive test per model state for each screening tool.**

| Test | State | Value (Range) | Description |
|------|-------|---------------|-------------|
| Nucleic acid amplification test (NAAT, Xpert MTB/RIF Ultra) | Susceptible | 0.006 (0.005 - 0.008) | (1 − specificity) for individuals in the community [49] |
| | Infected | 0.006 (0.005 - 0.008) | (1 − specificity) for individuals in the community [49] |
| | Cleared | 0.006 (0.005 - 0.008) | (1 − specificity) for individuals in the community [49] |
| | Recovered | 0.040 (0.020 - 0.060) | (1 − specificity) for individuals with a history of TB [57] |
| | Non-infectious TB | 0.044 (0.026 - 0.070) | (1 − specificity) for individuals with presumptive TB [47] |
| | Asymptomatic TB | 0.775 (0.676 - 0.856) | Sensitivity for smear-negative pulmonary TB in a care-seeking population [47] |
| | Symptomatic TB | 0.909 (0.862 - 0.947) | Sensitivity for pulmonary TB in a care-seeking population [47] |
| | Treated | 0.040 (0.020 - 0.060) | (1 − specificity) for individuals with a history of TB [57] |
| Chest radiography with CAD software interpretation (CXR) | Susceptible | 0.085 (0.069 - 0.134) | Proportion of individuals with abnormal CXR [5] |
| | Infected | 0.085 (0.069 - 0.134) | Proportion of individuals with abnormal CXR [56] |
| | Cleared | 0.085 (0.069 - 0.134) | Proportion of individuals with abnormal CXR [57] |
| | Recovered | 0.503 (0.481 - 0.524) | Proportion of individuals reporting previous TB treatment with abnormal CXR [58] |
| | Non-infectious TB | 0.677 (0.626 - 0.712) | Midpoint between values for Recovered/Treated and Symptomatic TB |
| | Asymptomatic TB | 0.677 (0.626 - 0.712) | Midpoint between values for Recovered/Treated and Symptomatic TB |
| | Symptomatic TB | 0.850 (0.770 - 0.900) | Sensitivity for bacteriologically confirmed TB [57] |
| | Treated | 0.503 (0.481 - 0.524) | Proportion of individuals reporting previous TB treatment with abnormal CXR [58] |

Probability of a positive test result for each screening diagnostic tool by model state. Probabilities were independently sampled from uniform distributions for each model run. CXR positivity values do not correspond to a specific CAD threshold and are explored further in the sensitivity analyses. Further descriptions on the probability of a positive test, including values for Xpert MTB/RIF and further investigation, are provided in **S3 Table**. CAD: Computer-aided diagnosis; TB: Tuberculosis.

outcomes throughout the results. Once the TB prevalence threshold was reached, we assumed the screening intervention would cease, and the model would revert to the BAU standard of care. We estimated the number of incident TB episodes and deaths averted compared with the BAU counterfactual. TB prevalence was defined as the sum of individuals with asymptomatic and symptomatic TB, while incident TB referred to the flow into symptomatic TB. To evaluate the performance of each screening algorithm, we also estimated the ratio of true positives (positive tests for non-infectious, asymptomatic, and symptomatic TB) to false positives (positive tests for non-disease states) treated. Additionally, we compared our model outputs with the results of the ACT3 trial by assessing the proportional reduction of TB prevalence after three rounds of community-wide screening with NAAT compared with BAU [16,34]. We set a time horizon of 2050 to balance recency with a sufficient duration for benefits to accrue.

## Cost outcomes

We took a simple and conservative provider approach to cost outcomes, broadly considering treatment and diagnosis in BAU and the various screening algorithms. To estimate the costs for the BAU passive detection counterfactual, we obtained cost estimates per individual from the NTP in Viet Nam (S4 Table). These included the average cost of TB treatment, covering expenditure for TB drugs, personnel, bacterial monitoring and overheads; it did not include costs associated to the management of adverse events due to treatment. The average cost of passive TB diagnosis accounted for the number needed to test, personnel costs, infrastructure, and actual test costs. All costs were categorised based on drug susceptibility, i.e., drug-susceptible or drug-resistant TB. Population-wide screening costs for each intervention algorithm were provided from the ACT3 trial and the ongoing ACT5 trial (S5 Table) [35]. These unit costs account for the number of tests performed, field staff, lab technicians, data managers, and consumables. For the main analysis, all cost estimates were independently sampled from gamma distributions, generated using the mean cost and a standard deviation of 20% of the mean value. All costs are presented in 2023 United States dollar (US$).

## Summary health outcomes

We estimated disability-adjusted life years (DALYs) averted in the simulated algorithms compared with the BAU baseline to quantify the health gains achieved by population-wide screening interventions (see details in S5 Text). Both DALYs due to symptomatic TB and post-TB sequelae were considered, using country-specific data on total DALYs in 2019 from Menzies et al. [31,35]. We calculated the estimated lifetime DALYs per person with TB by accounting for the number of incident TB in Viet Nam in 2019 [32]. We then used the weighted average age of individuals with TB in Viet Nam and estimated the proportion of the lifetime that would be lived from 2025 to 2050. We estimated that 6.6 DALYs (95%CI: 4.8-9.0) would be incurred per incident TB episode (including the resulting post-TB sequelae) in 2025, varying depending on the age at which they experience TB. No disability weights were applied for other disease or treatment states and transitions. DALYs were independently sampled from a uniform distribution of the ranges discounted at 3% per year [36]. We did not account for DALYs accrued during earlier disease states or due to adverse events associated to treatment.

## Summary cost-outcomes

We calculated the budget impact of each algorithm and the BAU counterfactual as the cumulative costs of treatment and screening/diagnosis for both BAU and the intervention up to 2050. The cost of front-loading was defined as the intervention-specific screening and treatment costs incurred, presented both as total cumulative costs and as annual averages over the intervention period. Additionally, we defined annual cost savings as the difference between BAU-specific diagnosis and treatment costs under the screening algorithms and the BAU counterfactual, averaged across the time horizon.

## Cost-effectiveness analysis

We calculated the incremental cost-effectiveness ratio (ICER) for each screening algorithm relative to the BAU counterfactual as well as in comparison with one another. The resulting ICERs were expressed as the cost per DALY averted associated with the screening intervention and these were evaluated in relation to an estimated cost-effectiveness threshold range of US$2,176 to US$3,283 in Viet Nam, based on a GDP per capita of US$3,817 in 2023 [37]. We excluded interventions from consideration based on simple dominance, where an alternative intervention was both more effective and less costly than the comparator, and extended dominance, where an alternative was more effective and more costly but offered better value for money (i.e., lower ICER) than the comparator [7].

## Sensitivity analyses

We performed several sensitivity analyses to test our model and assumptions. Firstly, we evaluated the cost effectiveness of reducing the unit price of NAAT cartridges from US$8 to US$1 (S5 Table); no price reduction was applied for CXR use. Secondly, for each algorithm we explored the number of rounds needed and the resulting impact on disease burden and cost-effectiveness of targeting two alternative TB prevalence thresholds: 100 or 20 per 100,000 people. Thirdly, we assessed the performance of using Xpert MTB/RIF instead of Xpert MTB/RIF Ultra in an alternative NAAT-only algorithm to evaluate the impact of a lower probability of a positive test on non-disease states (due to higher specificity) and disease states (due to lower sensitivity) (S3 Table). Fourthly, we evaluated the impact of further investigation for individuals who screened positive in each algorithm, assuming that in practice further evaluation by a clinician based on existing clinical, radiological, or microbiological information would be performed prior to initiating treatment. These measures are intended to guide appropriate treatment decisions, ensuring effective resource utilisation and minimising potential harm to individuals. In absence of clear data from the literature, we used performance of prolonged cough as a proxy (see S3 Table) and assumed a fixed cost of US$2 per; this value was illustrative and not intended to reflect the cost of any specific diagnostic test. Finally, we explored a revised CXR sensitivity scenario, assuming reduced test positivity for non-infectious and asymptomatic TB, equivalent to its performance in individuals who have recovered or been treated (**Table 1** and S3 Table).

## Results

### Model calibration

Model calibration generated 1,000 non-implausible parameter sets. **Fig 2** shows model runs with fitted parameters compared against calibration targets. The median and corresponding 95% uncertainty intervals (95%UI), calculated as the 2.5th to 97.5th percentiles, of the posterior parameter distributions, are presented in S2 Table.

### Rounds required to reach TB prevalence thresholds

At least six annual rounds of population-wide screening were needed to reach a TB prevalence of 50 per 100,000 people under the NAAT-only algorithm and eight for the CXR+NAAT algorithm (**Fig 3**, **Table 2** and S2 Fig). In contrast, under the CXR-only algorithm, treating everyone with radiological abnormalities without microbiological confirmation resulted

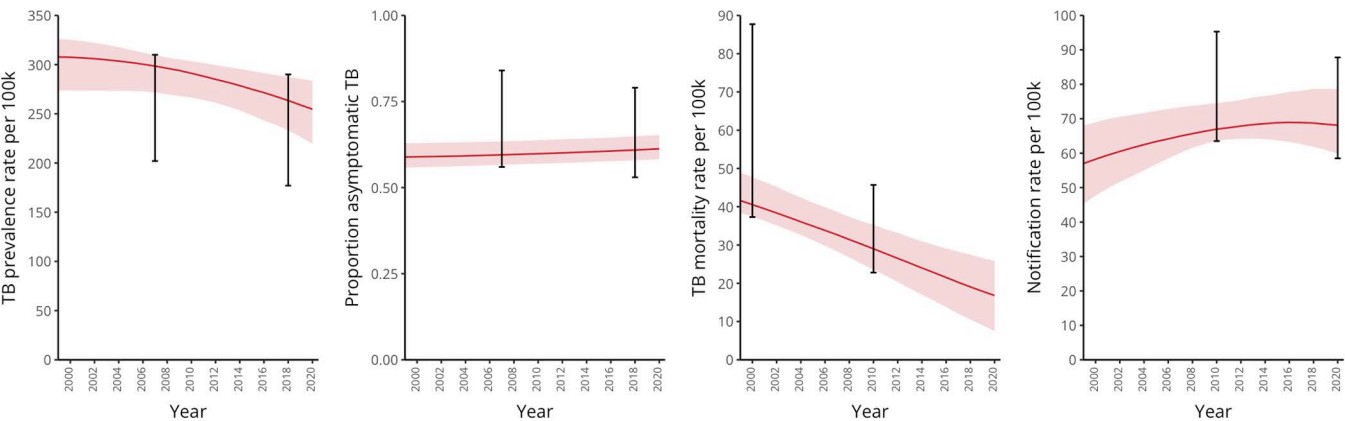

**Fig 2. Model outputs compared against calibration targets of TB in Viet Nam.** Fitted parameter sets were obtained by calibrating TB epidemiological data from Viet Nam using history matching with emulation. Red lines and shaded areas represent the median outputs and corresponding 95% uncertainty interval, respectively. Error bars indicate the time-specific calibration targets, with their values and sources specified in S1 Table.

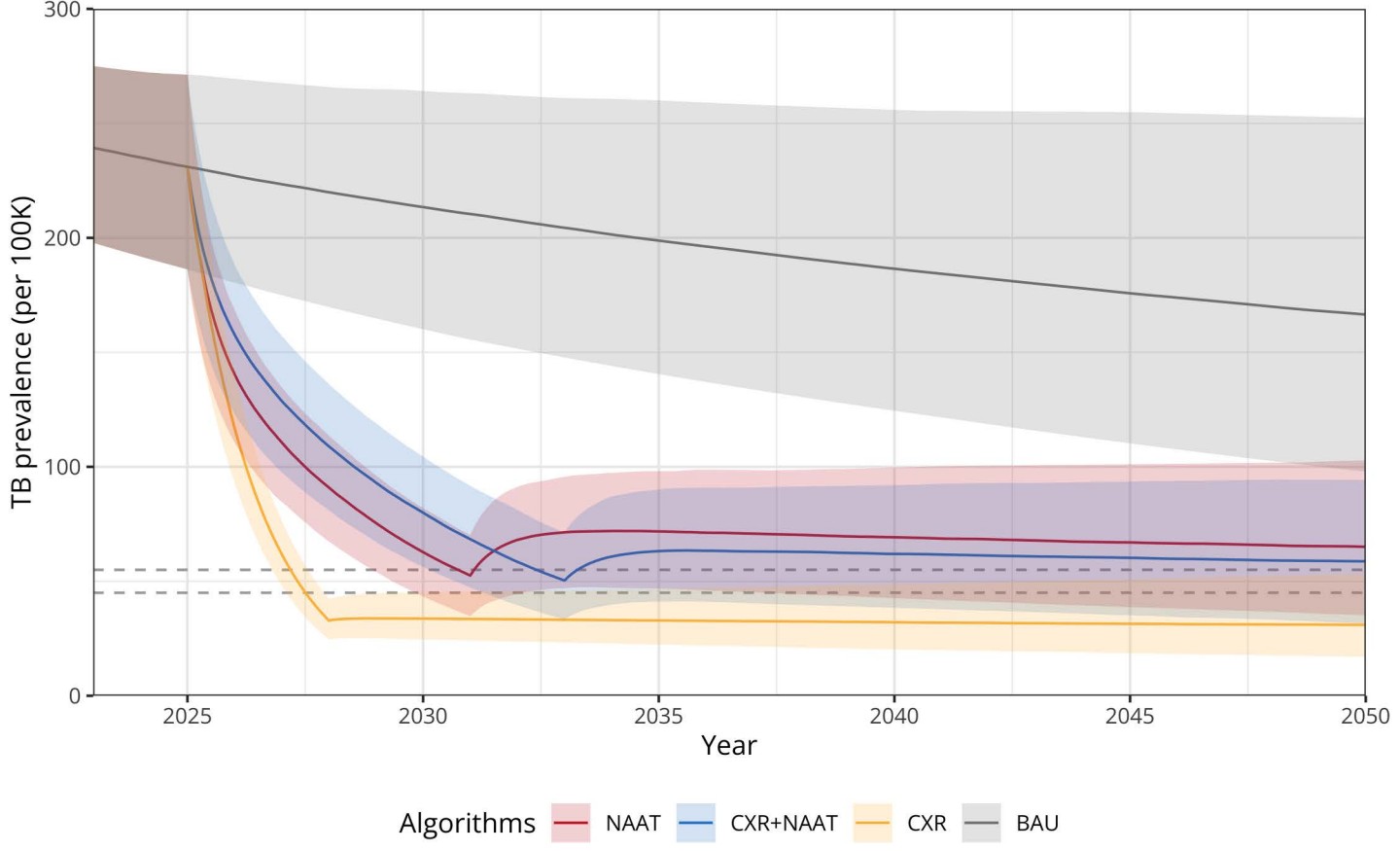

**Fig 3. TB prevalence reduction under population-wide screening algorithms.** TB prevalence over time under each population-wide screening algorithm until the TB prevalence threshold of 50 per 100,000 people is reached. Solid lines represent median TB prevalence, with shaded areas showing the lower (2.5% quantile) and upper (97.5% quantile) bounds. Dashed lines represent TB prevalence threshold (±10%). BAU: Business-as-usual; NAAT: Nucleic acid amplification test (Xpert MTB/RIF Ultra); CXR: Chest radiography with CAD software interpretation; CAD: Computer-aided diagnosis.

in a similar decline of TB prevalence after three annual rounds. Under BAU, the threshold was not reached within the time horizon. Additionally, we compared the performance of the NAAT-only algorithm, as simulated by our model, with the observed impact of the ACT3 clinical trial, which utilised the same screening algorithm. After three years of population-wide screening with the NAAT-only algorithm, we estimated a proportional reduction in TB prevalence of 58.7% (95%UI: 52.9-65.0), which sits between the reduction in prevalence seen in the intervention clusters and the comparison with the control clusters observed in the ACT3 trial of 67.7% (95%UI: 64.0-71.2) and 42.1 (95%UI: 36.7-47.1%), respectively (S3 Fig).

## Epidemiological impact of interventions

All screening algorithms resulted in lower cumulative TB incidence, TB deaths, and DALYs compared with BAU between 2025 and 2050 (Table 2). The cumulative TB incidence under both NAAT-based algorithms was similar, with 0.95 million (95%UI: 0.63-1.31) and 0.93 million (95%UI: 0.63-1.29) individuals projected to fall ill with TB under the NAAT and CXR+NAAT algorithms, respectively. Notably, TB incidence was further reduced under the CXR-only algorithm, with a cumulative TB incidence of 0.51 million (95%UI: 0.36-0.71). A similar trend was observed in cumulative TB deaths and DALYs (Table 2). Although implemented for only three annual rounds, the CXR-only algorithm correctly diagnosed 1.17 million

**Table 2. Performance of population-wide screening interventions to reach TB prevalence threshold of 50 per 100,000 people.**

| Screening algorithm | BAU | NAAT | | CXR+NAAT | | CXR |
|---|---|---|---|---|---|---|
| **Rounds required to reach threshold** | Not reached | 6 annual rounds | | 8 annual rounds | | 3 annual rounds |
| **Cumulative TB incidence** | 2.25m (95%UI: 1.57-3.04) | 0.95m (95%UI: 0.63-1.31) | | 0.93m (95%UI: 0.63-1.29) | | 0.51m (95%UI: 0.36-0.71) |
| **Cumulative TB deaths** | 273k (95%UI:123–475) | 104k (95%UI: 44–184) | | 99k (95%UI: 42–175) | | 60k (95%UI: 27–106) |
| **Cumulative DALYs** | 8.12m (95%UI: 5.85-10.83) | 3.74m (95%UI: 2.64-4.99) | | 3.85m (95%UI: 2.75-5.13) | | 2.21m (95%UI: 1.59-3.01) |
| **Cumulative TPs diagnosed through screening** | N/A | 555k (95%UI: 411–688) | | 578k (95%UI: 419–736) | | 1,165k (95%UI: 791-1,555) |
| **Cumulative FPs diagnosed through screening** | N/A | 2,779k (95%UI: 2,059-3,696) | | 1,717k (95%UI: 959-2,698) | | 31,619k (95%UI: 24,240–38,207) |
| **Unit price of NAAT** | N/A | US$8 | US$1 | US$8 | US$1 | N/A |
| **Cost of diagnosis/ screening** | 363m (95%UI: 222–578) | 2,428m (95%UI: 1,675-3,465) | 991m (95%UI: 693-1,360) | 1,211m (95%UI: 845-1,719) | 962m (95%UI: 679-1,335) | 350m (95%UI: 251–474) |
| **Cost of treatment** | 138m (95%UI: 86–209) | 336m (95%UI: 220–511) | | 253m (95%UI: 152–393) | | 2,609m (95%UI: 1,552–3,941) |
| **Budget impact** | 505m (95%UI: 328–757) | 2,766m (95%UI: 1,965-3,782) | 1,345m (95%UI: 999-1,755) | 1,478m (95%UI: 1,066-1,996) | 1,225m (95%UI: 894-1,614) | 2,954m (95%UI: 1,909-4,370) |
| **Annual cost of front-loading** | N/A | 427m (95%UI: 299–599) | 190m (95%UI: 138–258) | 161m (95%UI: 111–224) | 129m (95%UI: 92–174) | 949m (95%UI: 606-1,415) |
| **Annual cost savings** | N/A | 12.3m (95%UI: 6.5-21.4) | | 12.7m (95%UI: 6.7-21.4) | | 15.6m (95%UI: 9.2-24.7) |
| **ICER compared with BAU (US$ per DALY averted)** | N/A | 516 (95%UI: 233-1,073) | 189 (95%UI: 71–426) | 225 (95%UI: 85–520) | 167 (95%UI: 57–380) | 410 (95%UI: 178–929) |

Epidemiological performance and economic impact of population-wide screening interventions in Viet Nam by algorithm, conducted until the TB prevalence threshold of 50 per 100,000 people is reached. Values represent cumulative outcomes over a 25-year time horizon, extending up to 2050. Budget impact reflects the total cost of screening/diagnosis and treatment for both the intervention and BAU scenarios. The cost of front-loading refers to the average annual screening and treatment cost attributable to the intervention during the implementation period. Annual cost savings are calculated as the average annual difference in BAU-specific diagnosis and treatment costs between the intervention algorithm and the BAU counterfactual. BAU: Business-as-usual; CXR: Chest radiography; DALY: Disability-adjusted life year; FP: False positive; ICER: Incremental cost-effectiveness ratio; NAAT: Nucleic acid amplification test (Xpert MTB/RIF Ultra); TB: Tuberculosis; TP: True positive; UI: Uncertainty interval; US$: United States dollar.

(95%UI: 0.80-1.56) individuals with TB, with the majority having non-infectious TB (0.93 million; 95%UI: 0.60-1.31). However, this algorithm also resulted in a high number of diagnoses among non-disease individuals, with 31.6 million (95%UI: 24.2-38.2) individuals screened positive, yielding a true-positive (TP) to false positive (FP) ratio of 1:27. In contrast, the NAAT-based algorithms detected around half as many individuals with TB: 555 thousand (95%UI: 411–688) under the NAAT algorithm and 578 thousand (95%UI: 419–736) under the CXR+NAAT algorithm. The NAAT-only algorithm also diagnosed more FPs (2.8 million; 95%UI: 2.1-3.7) than TPs, resulting in a TP:FP ratio of 1:5. However, the two-step CXR+NAAT algorithm reduced the number of FPs overall (1.7 million; 95%UI: 1.0-2.7), leading to a TP:FP ratio of 1:3.

## Cost impact of interventions

Total intervention costs were substantial, with the CXR-only algorithm requiring US$3.0 billion (95%UI: 1.9-4.4), followed by the NAAT-only algorithm at US$2.8 billion (95%UI: 2.0-3.8) and the two-step CXR+NAAT algorithm at US$1.5 billion

(95%UI: 1.1-2.0) (Table 2). The cost of front-loading for the CXR-only algorithm was estimated at US$949 million (95%UI: 606-1,415) per year for the three years of the intervention, with projected annual cost savings of US$15.6 million (95%UI: 9.2-24.7) on average from 2025 to 2050. Similarly, the NAAT-only algorithm incurred US$427 million (95%UI: 299–599) in annual intervention-specific costs during the six years of intervention, resulting in average annual cost savings of US$12.3 million (95%UI: 6.5-21.4). The CXR+NAAT algorithm required US$1.3 billion (95%UI: 1.0-1.8) of intervention-specific costs, averaging US$161 million (95%UI: 111–224) per year for the first eight years, before becoming cost saving at US$12.7 million (95%UI: 6.7-21.4) annually compared to the BAU counterfactual (Table 2 and S4 Fig). Trends of annual cost savings suggest they will continue beyond the 25-year time horizon. For NAAT-based algorithms, the majority of costs were associated with screening procedures, accounting for 88.9% (95%UI: 81.6-93.2) of total costs in the NAAT-only algorithm and 84.2% (95%UI: 72.7-91.1) in the CXR+NAAT algorithm. In contrast, in the CXR-only algorithm, just 9.3% (95%UI: 5.4-15.8) of the total costs were related to screening, with the vast majority spent on treating individuals who screened positive.

## Cost-effectiveness analysis

Estimated ICERs against the BAU for all algorithms were below the estimated cost-effectiveness threshold range for Viet Nam. The CXR+NAAT algorithm averted 4.29 million DALYs (95%UI: 2.86-6.14) compared with BAU at an additional cost of US$967 million (95%UI: 523-1,488), giving an ICER of US$225 (95%UI: 85–520) per DALY averted. The CXR-only algorithm provided even greater effectiveness by averting an additional 1.61 million DALYs (95%UI: 0.86-2.56) compared with CXR+NAAT, but at double the total cost (US$2,954 million; 95%UI: 1,909-4,370), leading to an ICER of US$927 (95%UI: 393-1,124) per DALY averted compared with the CXR+NAAT algorithm (Table 2 and Table 3), which is also well below the cost-effectiveness threshold range. The NAAT-only algorithm was dominated by the other two alternatives, which were collectively cheaper and more effective.

## Sensitivity analyses

The performance of the algorithms under the different sensitivity analyses are shown in Table 2 and S6-S9 Tables. Firstly, lowering the unit price of NAAT to US$1 reduced screening costs, bringing the total costs to US$1.3 billion (95%UI: 1.0-1.8) and US$1.2 billion (95%UI: 0.9-1.6) in the NAAT-only and CXR+NAAT algorithms, respectively—an approximate reduction of 50% and 15% (Table 2). In terms of ICERs, this reduction was 63% for the NAAT-only algorithm and 26% for the CXR+NAAT algorithm (Table 2). In the cost-effectiveness analysis, the NAAT-only algorithm was still dominated

**Table 3. Cost-effectiveness of population-wide screening interventions for TB.**

| Screening algorithm | DALYs averted vs BAU | Additional costs (US$) vs BAU | Incremental DALYs | Incremental costs (US$) | ICER (US$ per DALY averted) |
|---|---|---|---|---|---|
| CXR+NAAT | 4.29m (95%UI: 2.86-6.14) | 0.97b (95%UI: 0.52-1.49) | 4.29m (95%UI: 2.86-6.14) | 0.97b (95%UI: 0.52-1.49) | 225 (95%UI: 85–520) |
| NAAT | 4.36m (95%UI: 3.09-6.23) | 2.25b (95%UI: 1.45-3.31) | Removed due to extended dominance with respect to CXR-only | | |
| CXR | 5.94m (95%UI: 4.18-7.97) | 2.44b (95%UI: 1.41-3.88) | 1.61m (95%UI: 0.86-2.56) | 1.49b (95%UI: 0.34-2.87) | 927 (95%UI: 393-1,124) |

Cost-effectiveness of population-wide screening interventions in Viet Nam by algorithm, conducted until the TB prevalence threshold of 50 per 100,000 people is reached. Values represent cumulative outcomes over a 25-year time horizon, extending up to 2050. BAU: Business-as-usual; CXR: Chest radiography; DALY: Disability-adjusted life year; ICER: Incremental cost-effectiveness ratio; NAAT: Nucleic acid amplification test (Xpert MTB/RIF Ultra); TB: Tuberculosis; UI: Uncertainty interval; US$: United States dollar.

 

despite this reduction in cartridge unit cost, with CXR-only remaining cost effective compared to all alternatives (S10 Table). Secondly, the relative epidemiological impact of the algorithms—assessed by cumulative TB incidence, TB deaths, and DALYs—remained consistent across different TB prevalence thresholds. However, the number of annual rounds required varied, ranging from two to three to achieve a prevalence of 100 per 100,000 and from four to twelve to achieve 20 per 100,000 (S6–S7 Tables). Thirdly, the use of Xpert MTB/RIF in an alternative NAAT-only algorithm was found to be less effective and more expensive than all three standard algorithms in the cost-effectiveness analysis (S10 Table). Fourthly, the addition of further investigation post-screening reduced the cumulative number of FPs across all algorithms, particularly in the CXR-only algorithm, where apparent FPs decreased by 90% compared to the main analysis (S8 Table). Given the reduced costs associated with the reduction in people treated, the CXR-only algorithm resulted in an ICER of US$113 (95%UI: 28–278) per DALY averted when compared to the BAU. For this analysis, CXR+NAAT averted more DALYs than CXR-only, at an ICER of US$1,293 (95%UI: 1,153-1,555) per DALY averted, while NAAT-only was marginally the most effective, but at a high additional cost, giving an ICER compared on CXR+NAAT well above the cost-effectiveness threshold range (S10 Table). Finally, with revised CXR sensitivity, the CXR+NAAT algorithm required one additional round to reach the 50 per 100,000 people TB prevalence threshold (S9 Table). Despite this, it only resulted in slight increase of the ICER compared to the BAU of US$283 (95%UI: 113–640) per DALY averted (S10 Table).

## Discussion

The study evaluates the implementation of various population-wide screening algorithms in order to achieve considerable reductions in TB prevalence in Viet Nam. While ambitious, these goals are aligned with the *End TB Strategy* targets [38], to which Viet Nam has committed. All three screening algorithms significantly reduced the TB burden compared to BAU, with the CXR-only algorithm achieving the greatest reductions in TB incidence, deaths, and DALYs. However, its high overtreatment rates made the two-step algorithm—combining CXR with a confirmatory NAAT—a more efficient option, averting a substantial number of DALYs at a relatively low cost. These outcomes are contingent on substantial front-loaded investments during the intervention, although reductions in NAAT cartridge costs can help offset these expenses. Ultimately, all interventions resulted in persistent annual cost savings compared to the BAU counterfactual up to 2050. Advocating for TB eradication on economic grounds is compelling [36], as trends suggesting that cost savings will persist alongside the long-lasting societal benefits of the intervention beyond the time horizon. The findings from this modelling exercise underscore the effectiveness of symptom-agnostic population-wide screening and provide guidance for implementing such large-scale strategies in high TB burden settings.

At face value, the CXR-only algorithm appears to be the most cost-effective option, achieving the greatest TB burden reduction with an ICER below the estimated cost-effectiveness threshold range [39]. However, several factors raise legitimate concerns about its suitability. Firstly, its epidemiological impact results in treating a large number of individuals with FP screens, which, while occurring in very few cases, carries risks such as serious adverse events, including hepatotoxicity [36], potentially offsetting the DALYs averted and leading to higher ICERs. Secondly, the CXR-only algorithm is the most expensive intervention over the 25-year time horizon, requiring substantial front-loaded investment of nearly US$1 billion annually during the initial years. While cost savings would be realised after the short intervention period, such funding demands in the medium-term could pose challenges. In contrast, the CXR+NAAT algorithm reduces the annual economic strain to US$160 million per year, albeit over a longer duration, ultimately resulting in a lower budget impact. Lastly, while the investment appears cost-effective within the estimated threshold range of US$2,176 to US$3,283 [16], Viet Nam has no official policy threshold, and judgements on what is affordable must therefore be made by national health decision-makers.

When comparing against the outcomes of the ACT3 trial, we observed that our model shows a similar impact in the proportional reduction of TB prevalence [40–42]. Given this, our model was able to extrapolate empirically validated methods for rapidly reducing TB burden to explore various algorithms and durations for population-wide screening interventions

in Viet Nam. However, our modelling focused on screening but did not explore other used measures that could be implemented on top of population-wide screening, such as TB preventive therapy for household contacts, social protection, or a generally more holistic approach that also addresses other structural determinants of TB [7]. Furthermore, our modelling assumed that after achieving intended TB prevalence thresholds, population-wide screening interventions are stopped, and TB care and prevention return to BAU. However, to sustain the momentum, alternative, more targeted approaches may then enable prevalence to decline to the level required for TB elimination, in accordance with the *End TB Strategy* targets [43]. Interventions such as contact and cluster investigations and targeted treatment of *Mtb* infection are likely to be more successful and more feasible in the context of the substantial reduction in both prevalence and incidence of TB once the threshold has been reached. Furthermore, we did not account for the catastrophic costs prevented, the economic contributions of individuals who remain healthy or alive, or the accumulating benefits beyond 2050 under BAU. Hence, the realistic long-term benefits of the intervention are likely strongly underestimated in this analysis.

Opting to use symptom-independent diagnostic methods helps to overcome the considerable limitations of conventional diagnostic and referral pathways. However, a challenge encountered relates to the diagnostic accuracy of these test—particularly specificity—in the context of population-wide screening interventions [44]. The cost-effectiveness of mass screening is linked to the ability of a test to accurately diagnose, as false positive diagnoses can drive up treatment costs, as illustrated by the CXR-only scenario [21,24]. This could largely be overcome by explicitly introducing further investigations post-screening as shown in our sensitivity analysis or by implementing confirmatory sputum culture for those testing positive on NAAT, coupled with ongoing surveillance for those who are negative on culture. In our analysis, this approach reduced FPs while maintaining much of the epidemiological benefit. As a case study, while our results indicate that CXR alone could rapidly reduce TB prevalence after a few annual rounds, this strategy is not feasible at present without an affordable, ideally non-sputum confirmatory test to establish the presence of viable *Mtb* as the cause of the pathology. Furthermore, the health consequences and social unacceptability of large-scale overtreatment would currently impede its recommendation. Lastly, the sensitivity and specificity values used in our model do not correspond to any particular CAD threshold. However, this remains a rapidly evolving field, and the optimal role of CXR—particularly in prevalence surveys and community-based screening—requires further evaluation [45].

A major strength of our study is that the natural history TB model used recognises earlier states of disease before symptomatic disease, adapting features of previously published models [16], and its use matches empirical data of community screening [23]. Encompassing the spectrum of TB within this framework provides insight into the impact of screening interventions according to disease state. This is not only the case for asymptomatic TB and its recognised contribution to transmission [45], but also that of non-infectious TB, a state where macroscopic evidence of disease can be detected and there is risk of further progression. An example of the large reservoir of non-infectious TB disease can be seen as the bounce back in prevalence once screening rounds stopped, reflecting the importance of detection and treatment beyond infectious TB disease [46]. Although our model was calibrated to Viet Nam TB epidemiology, the underlying framework and conclusions, particularly regarding the trade-offs between screening performance, intensity, and cost-effectiveness, are relevant to other high-burden settings. However, applying these findings elsewhere may require additional consideration of context-specific structural risk factors such as HIV.

Our study also has limitations. Firstly, our study highlighted how we have limited insight into test performance when used in communities. Estimates based on pooled diagnostic accuracy from clinical settings tend to underestimate the specificity of the test when applied to a community setting [47,48]; therefore, values for non-disease states were used based on the performance of the test in selected population studies [43,49]. Direct data from community screening would provide improved estimates of test positivity. Similarly, relying on care-seeking, clinic-based populations to inform NAAT sensitivity for symptomatic TB likely overestimates its performance in community-based screening and reflects an optimistic scenario [47]. In contrast, assuming independence between test results in the CXR+NAAT algorithm may underestimate sensitivity and thus the effectiveness of the algorithm, as individuals with higher bacillary burden are more likely to

test positive on both tests. In addition, the correlation of test results across screening rounds—as TB prevalence declines over time—may influence algorithm performance and merits further investigation. Moreover, the unknown distribution of sputum production across TB states highlights the need for sputum-independent diagnostic approaches, such as tongue swabs [50]. Sensitivity analyses, incorporating changes to test positivity or the addition of a further investigation step after screening, help to explore and validate the potential impact of these interventions. Secondly, our study did not account for the DALYs accrued due to overtreatment of the screened population as discussed above. Nonetheless, we are also likely underestimating the impact of screening on averting DALYs, as we only accounted for incident symptomatic TB, but did not look into the effect of these interventions on early TB disease diagnosis. Thirdly, our costing approach was simple in that we opted to obtain costs per individual screened and treated. Further granularity of the cost components would provide a deeper understanding of both the distribution of the intervention costs as well as that of costs averted under BAU. Additionally, our costing assumptions for conducting the interventions were centred around consumables and human resources but overlooked other sources of costs, including training and other scale-up activities prior to commencing the intervention. These costs are not negligible, especially in the start-up phase [51], and should be reflected for investment case for screening interventions. Fourthly, our model does not encompass or adjust based on key drivers of TB. Tuberculosis is a biosocial problem that unevenly impacts people of low socioeconomic status, likely given the greater exposure to many risk factors such as malnutrition, air pollution, and overcrowding [52,53]. Lastly, further exploration of post-TB respiratory disease is warranted. The number of TB survivors is immense, and proper assessment of healthcare-associated use and costs due to lingering morbidity of existing and not-prevented TB episodes under BAU needs to be considered especially when implementing large interventions [54].

Conventional symptom-centred, facility-based TB detection is insufficient [6]. Instead, proactive screening should complement existing routine passive detection. While Viet Nam currently implements active screening of limited high-risk populations, such as people living with HIV and close contacts of people with diagnosed TB, screening will need to extend beyond high-risk groups that only represent a limited fraction of prevalent TB. Sustained, multiple rounds will be required to ensure effectiveness [54,55]. Continuing the current strategy without significant enhancements is likely to result in a grave cost of inaction, causing TB to remain as serious a threat as it currently stands [55,56]. Instead, the resources required to sustain BAU care would be better invested in a short-term strategy of repeated population-wide screenings. Immediate efforts must focus upon overcoming initial logistical challenges to scaling-up, such as developing infrastructure and human resource capacity required to support population-wide screening. In parallel, efforts to enhance acceptability of large-scale screening by the community should be conducted. Prioritising population-wide screening will make significant strides towards reducing TB prevalence and would set an example for global TB elimination efforts. In conclusion, this modelling study has demonstrated pathways to rapidly reducing TB prevalence in Viet Nam, through population-wide screening. The cost of front-loading in these interventions promises to reduce morbidity and mortality and realise the *End TB Strategy* at a time when BAU is unlikely to reach the agreed targets.

## Supporting information

**S1 Text. Baseline model structure.**
(PDF)

**S2 Text. Baseline model equations.**
(PDF)

**S3 Text. Calibration methodology.**
(PDF)

**S4 Text. Screening model structure.**
(PDF)

**S5 Text. Disability-adjusted life years calculations.**
(PDF)

**S1 Fig. TB natural history model under population-wide screening.**
(PDF)

**S2 Fig. TB prevalence reduction under population-wide screening algorithms.**
(PDF)

**S3 Fig. Proportional reduction of TB prevalence under NAAT-only approach and ACT3.**
(PDF)

**S4 Fig. Incremental costs under CXR+NAAT algorithm.**
(PDF)

**S1 Table. Calibration targets.**
(PDF)

**S2 Table. Model parameter description, ranges, and non-implausible points.**
(PDF)

**S3 Table. Probability of a positive test per model state for each screening tool.**
(PDF)

**S4 Table. Costing estimates for business-as-usual TB diagnosis and treatment.**
(PDF)

**S5 Table. Costing estimates for population-wide screening algorithms.**
(PDF)

**S6 Table. Performance of population-wide screening interventions to reach TB prevalence threshold of 100 per 100,000 inhabitants.**
(PDF)

**S7 Table. Performance of population-wide screening interventions to reach TB prevalence threshold of 20 per 100,000 inhabitants.**
(PDF)

**S8 Table. Performance of population-wide screening interventions with further investigation post-screening.**
(PDF)

**S9 Table. Performance of population-wide screening interventions with revised CXR sensitivity.**
(PDF)

**S10 Table. Cost-effectiveness of population-wide screening interventions for TB.**
(PDF)

## Acknowledgments

The authors thank Dr Andrew Iskauskas for his valuable aid in the calibration of the model.

## Author contributions

**Conceptualization:** Guy B. Marks, Rein M.G.J. Houben.

**Data curation:** Alvaro Schwalb, Katherine C. Horton, Jon C. Emery, Martin J. Harker, Lara Goscé, Lara D. Veeken, Rein M.G.J. Houben.

**Formal analysis:** Alvaro Schwalb, Jon C. Emery.

**Investigation:** Alvaro Schwalb, Katherine C. Horton, Jon C. Emery, Martin J. Harker, Lara Goscé, Lara D. Veeken, Rein M.G.J. Houben.

**Methodology:** Alvaro Schwalb, Katherine C. Horton, Jon C. Emery, Martin J. Harker, Lara Goscé, Lara D. Veeken, Rein M.G.J. Houben.

**Supervision:** Katherine C. Horton, Guy B. Marks, Rein M.G.J. Houben.

**Writing – original draft:** Alvaro Schwalb.

**Writing – review & editing:** Katherine C. Horton, Jon C. Emery, Martin J. Harker, Lara Goscé, Lara D. Veeken, Frances L. Garden, Hai Viet Nguyen, Thu-Anh Nguyen, Khanh Luu Boi, Frank Cobelens, Greg J. Fox, Van Luong Dinh, Hoa Binh Nguyen, Guy B. Marks, Rein M.G.J. Houben.

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
