## [Decision Letter · Decision Letter 0]

27 Mar 2025

PGPH-D-25-00204

Potential impact, costs, and benefits of population-wide screening interventions for tuberculosis in Viet Nam: a mathematical modelling study

Dear Dr. Schwalb,

Thank you for submitting your manuscript to PLOS Global Public Health. After careful consideration, we feel that it has merit but does not fully meet PLOS Global Public Health’s publication criteria as it currently stands. Therefore, we invite you to submit a revised version of the manuscript that addresses the points raised during the review process.

We look forward to receiving your revised manuscript.

Kind regards,

Devan Jaganath

Academic Editor

Journal Requirements:

Additional Editor Comments (if provided):

Reviewers' comments:

Reviewer's Responses to Questions

**Comments to the Author**

1. Does this manuscript meet PLOS Global Public Health’s publication criteria ? Is the manuscript technically sound, and do the data support the conclusions? The manuscript must describe methodologically and ethically rigorous research with conclusions that are appropriately drawn based on the data presented.

Reviewer #1: Yes

Reviewer #2: Yes

2. Has the statistical analysis been performed appropriately and rigorously?

Reviewer #1: Yes

Reviewer #2: Yes

3. Have the authors made all data underlying the findings in their manuscript fully available (please refer to the Data Availability Statement at the start of the manuscript PDF file)?

Reviewer #1: Yes

Reviewer #2: Yes

4. Is the manuscript presented in an intelligible fashion and written in standard English?

Reviewer #1: Yes

Reviewer #2: Yes

5. Review Comments to the Author

Reviewer #1: The manuscript presents a well-executed mathematical modeling study evaluating the impact, costs, and benefits of population-wide tuberculosis (TB) screening interventions in Vietnam. The study is methodologically robust, clearly written, and provides valuable insights for policymakers considering large-scale TB screening programs. The results are well contextualized within existing literature, and the cost-effectiveness analysis is particularly relevant for decision-making.

The findings suggest that population-wide screening can significantly reduce TB prevalence, albeit with substantial upfront costs. The discussion appropriately balances the benefits of different screening approaches while acknowledging potential challenges such as high overtreatment rates in CXR-only algorithms.

The manuscript is suitable for publication in PLOS Global Public Health, pending minor revisions to enhance clarity and address a few methodological and interpretative concerns.

Major Strengths

Comprehensive Modeling Approach: The deterministic transmission model effectively captures TB epidemiology in Vietnam, incorporating various screening algorithms and their respective costs and benefits.

Policy-Relevant Cost-Effectiveness Analysis: The evaluation of incremental cost-effectiveness ratios (ICERs) provides practical insights into the feasibility of different screening strategies.

Comparison with ACT3 Trial Data: The validation of model outputs against real-world trial data strengthens the study’s credibility.

Consideration of Alternative Scenarios: The inclusion of sensitivity analyses, particularly regarding NAAT cost reduction and alternative prevalence thresholds, enhances the study’s applicability.

Minor Revisions

Clarification of Screening Algorithm Assumptions (Methods Section)

The description of screening eligibility criteria (p. 5, lines 143–145) could be slightly more explicit regarding whether individuals previously treated for TB are included in screening rounds.

It would be useful to specify whether the assumption that all infectious TB cases can provide sputum samples (p. 6, lines 147–148) is based on empirical data.

False Positive Rates and Overtreatment Implications (Results & Discussion)

The results section highlights a high false-positive rate in the CXR-only strategy (TP:FP ratio of 1:27, p. 9, lines 283–284). While the discussion acknowledges the ethical and financial implications, it may be beneficial to provide a brief estimate of potential adverse treatment effects (e.g., hepatotoxicity risks).

The discussion could further explore how confirmatory testing could be integrated to reduce overtreatment without undermining screening effectiveness.

Comparison with Other High-TB-Burden Countries

The study focuses on Vietnam, but given the global burden of TB, a short discussion on the generalizability of these findings to other high-burden settings (e.g., sub-Saharan Africa, India) would enhance the study’s impact.

Formatting and Minor Language Edits

Some figures and tables could benefit from clearer legends, particularly Table 1, where test performance parameters could be more explicitly described.

A few minor grammatical inconsistencies (e.g., "underscoring the importance of introducing proactive measures" could be "highlighting the need for proactive measures") should be addressed.

Recommendation

Accept with minor revisions

The manuscript is of high quality and makes a meaningful contribution to TB screening policy discussions. With minor clarifications, particularly regarding assumptions, overtreatment implications, and broader applicability, the paper will be ready for publication.

Reviewer #2: This is an important TB transmission-modeling analysis investigating the potential role of mass screening in achieving sustained reductions in TB disease burden. Strengths include the use of a natural history model that includes early TB states, the exploration of multiple diagnostic strategies, and the use of data from the ACT3/ACT5 trials for validation and cost estimation.

Overall, the analysis is well done and clearly presented, but I urge the authors to reconsider several of their diagnostic accuracy estimates, and to more fully explore uncertainty in these estimates and in natural history as it relates to the CXR-positive TB state. Specific comments are as follows:

Major:

1. My biggest concern is that this analysis overstates confidence of diagnostic accuracy estimates (particularly for the CXR-only screening) and makes several inaccurate assumptions about accuracy across different states (including for the algorithms that include NAAT). Specifically:

a) CXR positivity estimates are largely ungrounded in data. These will depend heavily on how TB is identified by CXR/CAD, yet the current value is just a midpoint between other estimates, and is assigned an overly narrow uncertainty range. I would like to see some consideration of what CAD thresholds are being considered (with 85% sensitivity for symptomatic TB and 91.5% specificity for all TB), and what proportion of the “unconfirmed” state those cutoffs are really likely to identify. I would also like to see more careful sensitivity analysis around this (with wider uncertainty).

b) NAAT positivity among the susceptible/infected/cleared is likely overstated. The estimate of specificity is from a population that included individuals with unconfirmed TB and recovered/treated TB among the TB-negatives – and it is likely that these groups accounted for most of the observed “false positive” Xpert Ultra results.

c) Sensitivity of NAAT for symptomatic TB is based on data from a care-seeking population, and the authors should instead use data from community screening studies or prevalence surveys to estimate the (lower) sensitivity among all symptom-screen-positive TB.

d) CXR positivity is assumed to be higher for unconfirmed TB than for treated TB, but the inverse could be true (that people who have progressed to symptomatic sputum+ TB and been treated may have persistent lung pathology and higher CAD scores than those with early sputum-negative TB.

e) It isn’t clear to me how NAAT prevalence among presumptive TB patients is relevant as a basis for estimating NAAT sensitivity in unconfirmed TB. Relevant data would focus on individuals with positive NAAT and negative culture, ideally from a community screening context, and with either longitudinal follow-up for outcomes off treatment or with other diagnostic testing (e.g. CT/PET) to determine presence of active TB.

f) Correlations:

- The assumption of test independence (i.e. multiplying individual probabilities of positivity for each test) will underestimate sensitivity of the two-step algorithm, because TB with higher sputum bacillary burden is also more likely to have abnormal x-rays. I would like to see the accuracy estimates account for this.

- This is tricker to represent in a compartmental modeling framework, but tests are also likely to be correlated over time, such that the same people may screen falsely positive or falsely negative on multiple rounds of screening (reducing impact but potential limiting overtreatment). This could be a discussion point.

g) The assumptions about sputum production seem implausible: Assming that every person with infectious TB can produce sputum will overestimate the impact of strategies that include NAAT, while assuming that only 60% of the TB-negative can produce sputum (considerably lower than in e.g. the TREATS study and others that tried to get a NAAT sample from everyone) will underestimate costs.

2. A major area of uncertainty, which isn’t adequately explored in current sensitivity analyses, is natural history as it relates to the unconfirmed state. For example:

a) How does the impact of the CXR intervention depend on the timing of the infected -> unconfirmed TB transition or the % of infected individuals who pass through an unconfirmed (Xray-positive sputum-negative) state?

b) For the rate of progression from unconfirmed to asymptomatic, the range considered is narrow, and the value isn’t really informed by the calibration process. I imagine that results depend heavily on this value?

c) The authors are defining the unconfirmed state in this analysis such that only a proportion have positive x-rays – but their earlier work to estimate related parameters used cohorts defined by x-ray findings. It is unclear how reliably those progression risks can be applied to the definitions they are using here.

3. The diagnostic delineation between different states should be more clearly described. E.g. is the line between unconfirmed and asymptomatic a positive sputum culture?

4. Considering only the median projections seems to underrepresent uncertainty of results. What is the rationale for this, and how much did the full posterior vary?

5. The secondary analysis modeling “further investigation” after a positive screen is poorly explained. After some thought, I can accept that clinicians might treat everyone with either a prolonged cough or positive confirmatory microbiological testing (e.g. culture), but this could be better explained. It’s also unclear why the cost is $2 per person, and whether this applies to all people who are screened (which seems high) or to only those who screened positive (in which case it is quite low given that the assumed 100% positivity among asymptomatic and symptomatic TB would require culture).

Minor comments:

6. The authors are trying to align with WHO terminology, but I wouldn’t use “unconfirmed” to describe sputum-negative active disease (previously called “minimal” or “noninfectious” TB disease). An important difference between the WHO framework (intended for programmatic relevance) and other recent classification schemes is that the WHO categories represent what tests *were done* and not what the results of more extensive testing *would have been*. Therefore, a lot of culture-positive TB may be “unconfirmed” in the WHO framework because it is NAAT-negative, or is sputum smear negative (where NAAT is unavailable), or is in a person who had trouble producing sputum and was diagnosed by x-ray. That is different from the state that precedes asymptomatic TB in this model, which is intended to represent sputum (culture) negative TB specifically. To align with WHO terminology but better represent biological states over diagnostic modality, I suggest “asymptomatic bacteriologically negative”, “asymptomatic bacteriologically positive”, and “symptomatic”.

7. Abstract: it’s odd to say “we designed” screening algorithms. Perhaps “simulated” or “represented”?

8. The “revised CXR sensitivity” assumptions should be explained briefly in the text before presenting the corresponding results; I could find them only in a footnote of a supplemental table.

6. PLOS authors have the option to publish the peer review history of their article (what does this mean? ). If published, this will include your full peer review and any attached files.

**Do you want your identity to be public for this peer review?** For information about this choice, including consent withdrawal, please see our Privacy Policy .

Reviewer #1: No

Reviewer #2: No

---

## [Decision Letter · Decision Letter 1]

4 Jul 2025

PGPH-D-25-00204R1

Potential impact, costs, and benefits of population-wide screening interventions for tuberculosis in Viet Nam: a mathematical modelling study

Dear Dr. Schwalb,

Thank you for submitting your manuscript to PLOS Global Public Health. After careful consideration, we feel that it has merit but does not fully meet PLOS Global Public Health’s publication criteria as it currently stands. Therefore, we invite you to submit a revised version of the manuscript that addresses the points raised during the review process.

Reviewer #2 has raised several questions regarding the analysis for you to consider and address.

We look forward to receiving your revised manuscript.

Kind regards,

Devan Jaganath

Academic Editor

Journal Requirements:

Reviewers' comments:

Reviewer's Responses to Questions

**Comments to the Author**

1. If the authors have adequately addressed your comments raised in a previous round of review and you feel that this manuscript is now acceptable for publication, you may indicate that here to bypass the “Comments to the Author” section, enter your conflict of interest statement in the “Confidential to Editor” section, and submit your "Accept" recommendation.

Reviewer #1: All comments have been addressed

Reviewer #2: (No Response)

2. Does this manuscript meet PLOS Global Public Health’s publication criteria ? Is the manuscript technically sound, and do the data support the conclusions? The manuscript must describe methodologically and ethically rigorous research with conclusions that are appropriately drawn based on the data presented.

Reviewer #1: Yes

Reviewer #2: Partly

3. Has the statistical analysis been performed appropriately and rigorously?

Reviewer #1: Yes

Reviewer #2: Yes

4. Have the authors made all data underlying the findings in their manuscript fully available (please refer to the Data Availability Statement at the start of the manuscript PDF file)?

Reviewer #1: Yes

Reviewer #2: Yes

5. Is the manuscript presented in an intelligible fashion and written in standard English?

Reviewer #1: Yes

Reviewer #2: Yes

6. Review Comments to the Author

Reviewer #1: Thank you for the opportunity to review this manuscript titled "Potential impact, costs, and benefits of population-wide screening interventions for tuberculosis in Viet Nam: a mathematical modelling study."

I have carefully reviewed the revised version and find that the authors have addressed previous comments thoroughly. The study is well-conducted, methodologically sound, and provides important insights into TB control strategies in Viet Nam and similar high-burden settings.

I recommend the manuscript for acceptance in its current form.

Reviewer #2: These minor revisions have improved the manuscript but haven’t addressed many of my concerns. I would like to see the authors engage more fully with the early-TB states they have defined and how diagnostic tests and algorithms would interact with them. Accuracy estimates for these TB states and in the screening context warrant revision in several places, along with clearer indications of where data are lacking and how influential that uncertainty might be.

The following are the revisions that, in my opinion, would be most critical:

1. In response to comments by both reviewers about sputum production, the authors cite Kampala and ACT3 screening studies, but neither of those studies support the assumption that sputum production differs by TB status or that everyone with TB can produce sputum. These parameter values warrant more careful consideration and revision.

2. Authors respond to questions about their CXR accuracy estimates by saying they don’t correspond to any particular CAD cutoff, but they don’t address the more fundamental question of whether *any* cutoff would simultaneously have the combination of sensitivity and specificity values that they chose. As the authors used prevalence surveys as the basis for their CXR specificity estimate, and most prevalence surveys did both CXR and microbiological testing for everyone with symptoms, it should be possible to estimate sensitivity of CXR (using the same diagnostic cutoff) in those same surveys.

3. While it is true data are sparse for estimating the sensitivity of NAAT in a screening context, the authors should at a minimum make clear in Table 1 that they are estimating sensitivity based on care-seeking populations.

4. Re: correlations, I would like to see the authors revise their sensitivity and specificity estimates for the CXR + NAAT algorithm to account for expected test non-independence. (The correlations of an individual’s test results over time between different screening rounds are the aspect of this that I think can be a discussion point.)

5. Authors are inconsistent re: the symptom status of “non-infectious TB”. They state in the methods that these are “without bacteriological evidence of TB disease, infectiousness, *or symptoms*”, which I think is the appropriate definition based on their model’s structure and how they estimated natural history parameters. But they contradict this choice when trying to justify their use of presumptive-TB patients to estimate NAAT sensitivity in unconfirmed TB. Ther are three problems with using presumptive-TB prevalence to estimate NAAT sensitivity:

a. They (rightly, for their model structure) define their “noninfectious” TB state as asymptomatic, which presumptive TB patients are not.

b. Presumptive-TB patients include many people with TB and higher bacilary burdens than the noninfectious-TB population.

c. More fundamentally, there just isn’t any direct relationship between the prevalence of NAAT positivity among presumptive-TB patients (a mix of people with symptomatic bacteriologically positive TB, symptomatic bacteriologically negative TB, and no TB) and the sensitivity (prevalence of NAAT positivity) among people with [symptomatic] bacteriologically negative TB.

If there are no better data that can be used to estimate this parameter, then that high uncertainty should be made more explicit and explored more fully in sensitivity analyses.

6. I would still like to see sensitivity analysis related to assumptions of the underlying natural history model. I understand how the underlying state-transition parameters were estimated in previous work and how the screening intervention was applied simultaneously across states (and I apologize if my previous comment was not worded clearly), but I would like to see the sensitivity of the impact of repeated screening to the values of the transition rate from infected to noninfectious-TB, and to the proportion of TB that passes through the noninfectious state.

7. PLOS authors have the option to publish the peer review history of their article (what does this mean? ). If published, this will include your full peer review and any attached files.

**Do you want your identity to be public for this peer review?** For information about this choice, including consent withdrawal, please see our Privacy Policy .

Reviewer #1: No

Reviewer #2: No

---

## [Editor Report · Decision Letter 2]

28 Jul 2025

Potential impact, costs, and benefits of population-wide screening interventions for tuberculosis in Viet Nam: a mathematical modelling study

PGPH-D-25-00204R2

Dear Dr. Schwalb,

We are pleased to inform you that your manuscript 'Potential impact, costs, and benefits of population-wide screening interventions for tuberculosis in Viet Nam: a mathematical modelling study' has been provisionally accepted for publication in PLOS Global Public Health.

Best regards,

Devan Jaganath

Academic Editor